# Pushdown Layers: Encoding Recursive Structure in Transformer Language Models

**Shikhar Murty**[†]  **Pratyusha Sharma**[‡]  **Jacob Andreas**[‡]  **Christopher D. Manning**[†]

[†]Computer Science Department, Stanford University   [‡]MIT CSAIL

{smurty, manning}@cs.stanford.edu, {pratyusha, jda}@mit.edu

## Abstract

Recursion is a prominent feature of human language, and fundamentally challenging for self-attention due to the lack of an explicit recursive-state tracking mechanism. Consequently, Transformer language models poorly capture long-tail recursive structure and exhibit sample-inefficient syntactic generalization. This work introduces *Pushdown Layers*, a new self-attention layer that models recursive state via a *stack tape* that tracks estimated depths of every token in an incremental parse of the observed prefix. Transformer LMs with Pushdown Layers are syntactic language models that autoregressively and synchronously update this stack tape as they predict new tokens, in turn using the stack tape to softly modulate attention over tokens—for instance, learning to "skip" over closed constituents. When trained on a corpus of strings annotated with silver constituency parses, Transformers equipped with Pushdown Layers achieve dramatically better and 3-5x more sample-efficient syntactic generalization, while maintaining similar perplexities. Pushdown Layers are a drop-in replacement for standard self-attention. We illustrate this by finetuning GPT2-medium with Pushdown Layers on an automatically parsed WikiText-103, leading to improvements on several GLUE text classification tasks.

## 1 Introduction

An important property of human language and thought is *recursion*, which allows us to compose and reason about complex objects in terms of simpler constituents (Hauser et al., 2002). While extensively studied in natural language syntax and semantics, recursion is also a key component of several other aspects of intelligent behaviors including mathematical reasoning, programming, and goal-directed planning. Most recursion-capable systems model recursive processes via a stack memory, which is updated as new computation is performed.

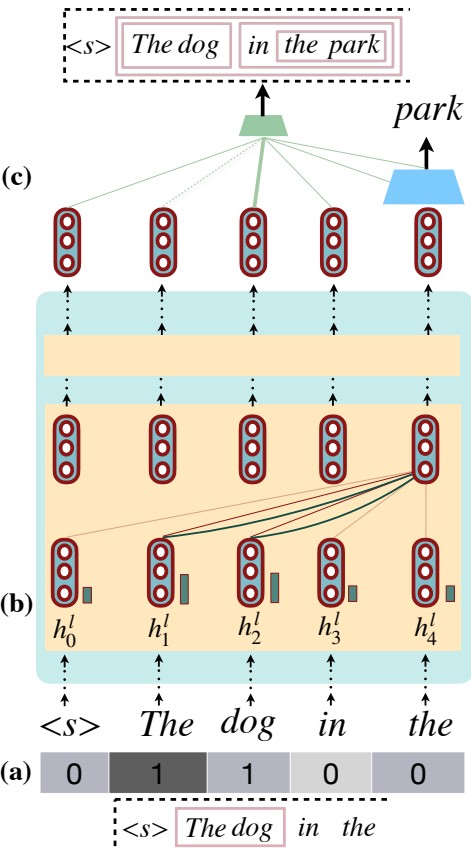

Figure 1: (a) Pushdown Layers use a stack-tape to featurize contents of an explicit stack, in terms of estimated token depths, where the stack represents incremental parses. (b) These depths map onto depth embeddings (in blue) that are added to token keys before computing attention scores, softly biasing attention towards a recursive syntactic computation. (c) The stack is updated *synchronously* with the newly predicted word, via an attachment head that selects a constituent to reduce the newly predicted word with, via attention.

For instance, a programming language may implement recursion by maintaining a run-time stack of caller-callee frames, storing intermediate outputs in the stack, and updating the stack as new function calls are made. Similarly, a shift-reduce parser implements recursion through a stack of intermedi-

ate constituents, shifting tokens onto the stack as they are observed, and occasionally reducing stack elements into constituents as they are completed.

In contrast, the self-attention mechanism underlying modern neural sequence models has no explicit mechanism to maintain a stack memory as it generates strings, and instead relies on hidden representations to implicitly but imperfectly encode such information (Manning et al., 2020). While this encoding can model bounded recursive structure in formal languages (Yao et al., 2021), it is unclear if it is sufficient for robust syntactic generalization, especially under data-constrained settings.

In this work, we show that an explicit stack memory mechanism can improve syntactic generalization in Transformer language models (LMs). We introduce *Pushdown Layers*[1], a drop-in replacement for standard self-attention that augments Transformer LMs with stack memory. This memory is modeled using a *stack tape* that stores estimated depths of every token in an incremental parse of the observed prefix. The stack tape is updated autoregressively: as new tokens are predicted, Transformers with Pushdown Layers (Pushdown Transformers) synchronously make probabilistic *attachment decisions* to either "shift", thus assigning the newly predicted token a depth of 0, or "reduce" with one of the constituents in the prefix so far, updating token depths accordingly (see Fig. 1). This stack tape is used to additively and softly modulate the attention of the Transformer over tokens—for instance, Pushdown Layers may guide the LM to only attend to head words of constituents, or skip over reduced constituents by decreasing attention.

Pushdown Transformer LMs are *syntactic language models* that learn joint probabilities of sequences and parses in terms of individual word predictions and structure-building operations, and can be trained on any text corpus annotated with constituency parses. But unlike other syntactic language models with structural supervision (Vinyals et al., 2015; Choe and Charniak, 2016; Qian et al., 2021; Sartran et al., 2022), Pushdown Layers do not change the output space of the underlying sequence model, and impose no constraints on attention mechanisms—the manner in which Pushdown Layers use syntactic structure for representation building is learnt purely via gradient descent.

Pushdown Transformers obtain strong general-

---

[1]We borrow this term from pushdown automata, which are finite state machines augmented with stacks.

ization improvements over standard Transformer LMs. When trained on depth-bounded Dyck strings and evaluated on deeper Dyck strings, Pushdown Transformers improve performance over baseline LMs by over 25% (Section 4.1). When trained on sentence-level language modeling on the BLLIP-LG datasets of Hu et al. (2020), Pushdown Transformers improve syntactic generalization over standard Transformer LMs by 5–13 points as well as other joint models of strings and parses such as Qian et al. (2021); Sartran et al. (2022) by 0.3–4 points (Section 4.2). When trained on a new, 100-million-token dataset of parsed Wikipedia articles we call WIKITREES, Pushdown Transformers match the syntactic generalization of ordinary Transformers with 3–5× less data. Finally, when Pushdown Layers are inserted into a pre-trained GPT-2 (medium) model and fine-tuned on WIKITREES they yield improvements of 0.3–1 points on several GLUE text classification tasks.

## 2 Background

**Multi-Head Self-Attention.** Transformer language models (Vaswani et al., 2017) are a class of neural sequence models that use multi-head *self-attention* to obtain contextualized representations of tokens in a sequence, which are then used to predict the next token. In particular, let $x = \{x_1, x_2, \ldots, x_n\}$ be an input sequence. Let $\boldsymbol{h}_i^l \in \mathbb{R}^d$ be the hidden representation of the $i^{\text{th}}$ token at the $l^{\text{th}}$ attention block. Then, the hidden representation of the $i^{\text{th}}$ token is updated as

$$\boldsymbol{h}_i^{l+1} = \text{FF}(O \cdot [\text{A}_1(\boldsymbol{h}_{\leq i}^l), , \cdots, \text{A}_K(\boldsymbol{h}_{\leq i}^l)]), \tag{1}$$

where $O \in \mathbb{R}^{d \times d}$ is a learnt matrix, FF denotes a feed-forward + residual + layer-norm block, and $\text{A}_p$ is the $p^{\text{th}}$ self-attention head. Each attention head performs a weighted average over its inputs,

$$\text{A}_p(\boldsymbol{h}_{\leq i}^l) = \sum_{j=1}^{i} \alpha_{ij} W_{\text{value}}^p \boldsymbol{h}_j^l, \tag{2}$$

where $\alpha_{ij}$ is the *attention weight* assigned to the $j^{\text{th}}$ token by the $i^{\text{th}}$ token. These attention weights are computed as

$$\alpha_{ij} = \text{softmax}[(W_{\text{key}}^p \boldsymbol{h}_j^l)^\top W_{\text{query}}^p \boldsymbol{h}_i^l]. \tag{3}$$

Each self-attention head introduces learnt parameters $W_{\text{key}}^p, W_{\text{query}}^p, W_{\text{value}}^p \in \mathbb{R}^{d/K \times d}$.

**Limitations of Self-Attention.** When trained on text corpora, transformers implicitly encode several aspects of linguistic structure unsupervisedly (Clark et al., 2019; Hewitt and Manning, 2019; Murty et al., 2023). However, there is mounting evidence that recursion, a key feature of human language, remains a challenge. Hahn (2020) shows theoretically that hard-attention cannot model simple recursive structures like 2DYCK (see Section 6 for an extended discussion). Empirically, Lakretz et al. (2022) show that self-attention struggles on center embedding phenomenon, and Zhang et al. (2023) show poor performance on simple recursive tree-traversal problems. We hypothesize that a key reason for poor modeling of recursive structure in self-attention is a lack of an explicit structural inductive bias. One common way to add such an inductive bias is via joint modeling of strings and syntactic structure, which we introduce next.

**Syntactic Language Models.** Let $y$ be the ground-truth syntactic parse of $x$. A long line of work (Vinyals et al., 2015; Dyer et al., 2016; Choe and Charniak, 2016; Qian et al., 2021; Sartran et al., 2022) considers learning joint distributions $p(x, y)$ to incorporate explicit syntactic structure into neural language models, by learning to output a sequence of *transition actions*,

$$p(x, y) = p(\mathbf{a}_{xy}) = \prod_i p(a_i \mid a_{<i}) \qquad (4)$$

where actions $a_i$ correspond to both word-level predictions as well as *structural actions* corresponding to opening and closing of constituents, building up the parse tree in a top-down, left-to-right manner. Recent work explores using Transformers to parameterize these joint distributions. For instance, Qian et al. (2021); Sartran et al. (2022) train Transformer LMs over transition actions (Parsing as Language Modeling or PLM), sometimes with constrained attention heads (PLM-mask), and Transformer Grammars (TG; Sartran et al., 2022) model transition actions with Transformers, also with hard constraints on attention to model shift/reduce actions.

These models have several limitations that motivate our proposed approach. First, their outputs are sequences of transition actions that include both text and tree-building operations; as each constituent in a parse tree has an opening and closing transition action, and there are $\approx n$ constituents for $x$, this increases input length by a factor of 3, leading to significant computation and memory over-

head. Second, inference in neural models operating on transitions require bespoke decoding procedures that carefully balance tradeoffs between high-entropy word-level predictions and low-entropy structural predictions (Stern et al., 2017). Finally, to explicitly bias Transformer computations to mirror the recursive structure of parse trees, some approaches like PLM-mask (Qian et al., 2021) and TGs (Sartran et al., 2022) impose hard constraints on attention patterns. Pushdown Layers provide a softer syntactic bias that is amenable to gradient-based learning, while having broader applicability to phenomena beyond local tree-structuredness, such as topical dependencies, coreference, etc.

## 3 Pushdown Layers

Transformer LMs with *Pushdown Layers* are syntactic language models that generate strings while simultaneously building a parse tree over these strings from left to right. This parse tree is built incrementally by tracking the recursive state of every token, which is synchronously updated along with word-level predictions. This recursive state is represented via our *stack tape* as tree-depths of every prefix token, and updates are realized with a stack. The contents of the stack tape are used to *softly modulate* attention over prefix tokens via additive offsets to attention logits (Fig. 2).

### 3.1 Stack Tape

Like ordinary self-attention, Pushdown Layers take a sequence of hidden states $\{\boldsymbol{h}_k^l\}$ as input, and output a sequence $\{\boldsymbol{h}_k^{l+1}\}$. Additionally, Pushdown Layers use a *stack tape* $\mathcal{W}_k \in \{0, k\}^k$ to simulate a pushdown automaton that performs shift/reduce operations over tokens as they are predicted (Fig. 2). The contents of the stack tape encode recursive state by tracking the depth of each token within reduced constituents in the stack. Concretely, after observing the prefix $x_{\leq k} = \{x_1, x_2, \ldots, x_k\}$, $\mathcal{W}_k[j] = 0$ if token $x_j$ has not been reduced with any other token, while $\mathcal{W}_k[j] = p$ means that $x_j$ has appeared in $p$ reduce operations such that the resulting *constituent* has token $x_j$ at depth $p$—in Fig. 2, the stack tape encodes [1, 1, 0] for the incremental parse [*The dog*] *is*.

**Updating the Stack Tape.** As shown in Fig. 2, along with predicting the next word *happy*, Transformers with Pushdown Layers (*Pushdown Transformers*) make an attachment decision to update their stack tape. In our running example, this is

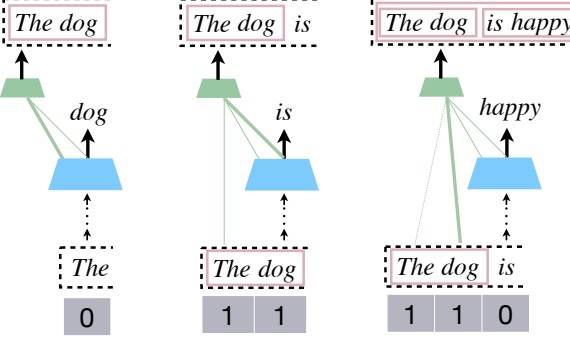

Figure 2: Illustration of how the parse [[*The dog* [*is happy*]] is built as a unique sequence of stack-tape updates in Pushdown LMs. Here, as the word *happy* is predicted, the attachment head chooses a constituent (bolded) from the current incremental parse, via attention. Attachment decisions are made to constituents by attending to their rightmost token, and none of the other tokens of a constituent can be attended to (shown as dashed lines). These attachment decisions are used to update depth values in the tape.

done by selecting a constituent from the incremental parse [*The dog*] *is happy*.

Concretely, given prefix $x_{<k}$, Pushdown Transformers predict the next token $x_k$ as well as an update to the stack tape $\mathcal{W}_{k-1}$. This is done by selecting a token $r_k$ to reduce with, out of candidate tokens $\{x_1, x_2, \ldots, x_k\}$, via attention over hidden states $\{\boldsymbol{h}_1^L, \boldsymbol{h}_2^L, \ldots, \boldsymbol{h}_{k-1}^L, \tilde{\boldsymbol{h}}_k^L\}$, where $L$ is the final layer of the Transformer, and $\tilde{\boldsymbol{h}}_k^L$ is a vector representation for the newly predicted token $x_k$, obtained as $\tilde{\boldsymbol{h}}_k^L = \mathrm{MLP}(x_k, \boldsymbol{h}_{k-1}^L)$. This vector attends to all tokens to make a probabilistic attachment decision,

$$p(r_k = j \mid x_{<k}; \mathcal{W}_{k-1}) \propto$$
$$\begin{cases} (\boldsymbol{h}_j^{L\top} W^\top \tilde{\boldsymbol{h}}_k^L) & \text{if } j \neq k,\ \textit{shift + reduce} \\ (\tilde{\boldsymbol{h}}_k^{L\top} W^\top \tilde{\boldsymbol{h}}_k^L) & \textit{shift only} \end{cases} \quad (5)$$

where $W \in \mathbb{R}^{d \times d}$ is a learnt parameter matrix. We use these probabilities to select token $r_k = \arg\max p(j \mid x_{<k}; \mathcal{W}_{k-1})$ to reduce $x_k$ with, and the stack tape is updated accordingly via Algorithm 1. Note that attachment decisions to constituents are made by computing the attachment score for the rightmost token in the constituent. In our running example, the model selects the constituent [*The dog*] by selecting the word *dog*, forming the parse [[*The dog*] [*is happy*]] and updating the stack tape from [1, 1, 0] → [2, 2, 2, 2].

---

**Algorithm 1:** Stack Tape Update

**Input**: $\mathcal{W}_{k-1}$, $k$, $r_k$, stack
**Output**: $\mathcal{W}_k$, stack

**UpdateStackTape**($\mathcal{W}_{k-1}$, $k$, $r_k$, stack)

$\mathcal{W}_k \leftarrow \mathcal{W}_{k-1}$
constituent $\leftarrow [k]$
**if** $r_k == k$ **then**
$\quad$ stack.push(constituent)
$\quad$ return
**end**
**while** True **do**
$\quad$ top $\leftarrow$ stack.pop()
$\quad$ // *Perform a reduce*
$\quad$ constituent $\leftarrow$ top + constituent
$\quad$ // *Update depths in stack tape*
$\quad$ **forall** $d \in$ constituent **do**
$\quad\quad$ $\mathcal{W}_k[d]$ += 1
$\quad$ **end**
$\quad$ **if** top == $r_k$ **then**
$\quad\quad$ break
$\quad$ **end**
**end**
stack.push(constituent)

---

### 3.2 Computing Attention Scores

We map contents of $\mathcal{W}_k$ onto a *per-layer* depth embedding $\boldsymbol{d}_{kj}^l$ for every token $j \in \{0, 1, \ldots, k\}$. These depth embeddings are added to attention keys, resulting in a *locally additive* modulation to attention scores,

$$\tilde{\alpha}_{kj}^l = \mathrm{softmax}\big([\boldsymbol{h}_j^l + \boldsymbol{d}_{kj}^l]^\top W_{\text{key}}^\top W_{\text{query}} \boldsymbol{h}_k^l\big). \tag{6}$$

Of course, since these logits are themselves part of a softmax and non-linearities, the overall effect can be arbitrarily non-linear. These modified attention weights are used to compute contextualized vectors using Eq 2 and Eq 1.

### 3.3 Training and Inference

**Training.** Given a corpus of strings annotated with parses, we first extract ground-truth values of $\mathcal{W}_k$ for every prefix $x_{\le k}$. We also extract ground-truth attachment decisions for $x_k$, given prefix $x_{<k}$. With these quantities precomputed, we can train Pushdown Transformers in *parallel*, like standard Transformers. Attachment probabilities (Eq 5) are supervised with ground-truth attachments, along with the standard LM objective, all using hidden

states that are contextualized using the Pushdown Layer attention mechanism that uses the precomputed stack tape.

**Inference.** For any string $x$ and parse $y$, joint probability $p(x, y)$ factorizes as a product of word-level and attachment scores as

$$p(x,y) = \prod_{k=1}^{n} \Big( p(x_k \mid x_{<k}; \mathcal{W}_{k-1}) \times$$
$$p(r_k \mid x_{<k}; \mathcal{W}_{k-1}) \Big). \quad (7)$$

While computing the full marginal $p(x) = \sum_y p(x, y)$ is computationally infeasible due to the large space of possible parses, we approximate this by marginalizing over a smaller subset with beam search. Crucially, since our model predicts words and structural actions in *parallel* rather than sequentially, we do not need to use complex word-synchronous decoding procedures (Stern et al., 2017) that introduce additional hyperparameters.

### 3.4 Implementation Details

**FLOPs and memory overhead.** Consider query and key matrices $Q \in \mathbb{R}^{n_d \times d}, K \in \mathbb{R}^{n_s \times d}$ where $n_d$ and $n_s$ refer to destination (hidden states attending) and source (hidden states being attended to). Let $S \in \mathbb{R}^{n_d \times n_s}$ be the (lower-triangular) matrix denoting pre-computed stack tape values for every prefix. For each Pushdown Layer, we use $S$ to index into depth embeddings to obtain $D \in \mathbb{R}^{n_d \times n_s \times d}$, which is added to $K$ to obtain $K_D \in \mathbb{R}^{n_d \times n_s \times d}$. Unlike standard self-attention which multiplies $Q$ and $K$ directly, Pushdown Layers multiply $Q$ (a 2D tensor) with $K_D$ (a 3D tensor). This is done by casting $Q$ into a 3D tensor $\in \mathbb{R}^{n_d \times 1 \times d}$ and performing a batched matrix multiplication with $K_D$, leading to the same number of operations as standard self-attention [2]. However, since Pushdown Layers require storing 3D tensors for keys, this increases memory requirements from $O(n_d \cdot n_s + n_s \cdot d + n_d \cdot d)$ to $O(n_d \cdot n_s + n_s \cdot n_d \cdot d + n_d \cdot d)$. We provide standalone code for implementing a Pushdown Layer block in Appendix D.

**Attending to hidden states with old memory.** Pushdown Transformers build parse trees incrementally from left-to-right, and so, depth values of prefix tokens change as new tokens are predicted.

Thus, a token at position $i$ builds its representation based on attending to $x_{\leq i}$ with a stack tape that may soon become "stale" due to future transition operations that reduce tokens in $x_{\leq i}$ with new tokens. As an example, suppose we have the incremental parse [[*The dog*] [*in* [*the park*]]]. Here, the representation for *in* attends to representations of *The*, *dog* and *in* with depths [1, 1, 0] while the representation for *park* attends to these representations with *updated* depths [2, 2, 2].

## 4 Experiments

### 4.1 Warm-up: Dyck Languages

We train 6 layer LMs with Pushdown Layers (Pushdown-LM) as well as standard LMs on 100k strings sampled from $\text{DYCK}_{20,10}$, the language of well-nested brackets with 20 bracket types and max-nesting depth of 10. To ensure that improvements are not merely due to multi-task learning with an attachment head, base-LM is also trained with an attachment loss in a standard multi-task learning setup. To test generalization, models are provided an input prefix from a separate DYCK language, and evaluated on choosing the correct closing bracket. Specifically, we test generalization to DYCK strings with deeper nesting of 15–50, and DYCK strings with longer-range dependencies than seen at training time (measured as the distance to the matching bracket that needs to be closed). From Table 1, we find that Pushdown-LM obtains over 25% accuracy point improvement over standard language models at generalizing to deeper structure, as well as large improvements at generalizing to longer-range dependencies.

| | Base-LM | Pushdown-LM |
|---|---|---|
| **Long-Range Dependencies** | | |
| DYCK (50) | 90.0 | **96.5** |
| DYCK (100) | 81.0 | **88.0** |
| DYCK (200) | 40.6 | **61.2** |
| DYCK (300) | 14.1 | **42.9** |
| **Deeper Embedded Structure** | | |
| Depth Gen. | 40.6 | **68.3** |

Table 1: Evaluating LMs at closing Dyck prefixes with longer dependencies (dep. length in brackets) and deeper structure. We find significant improvements from using Pushdown Layers over standard self-attention.

---

[2] We note that standard self-attention is faster in practice due to better GPU memory bandwidth management,

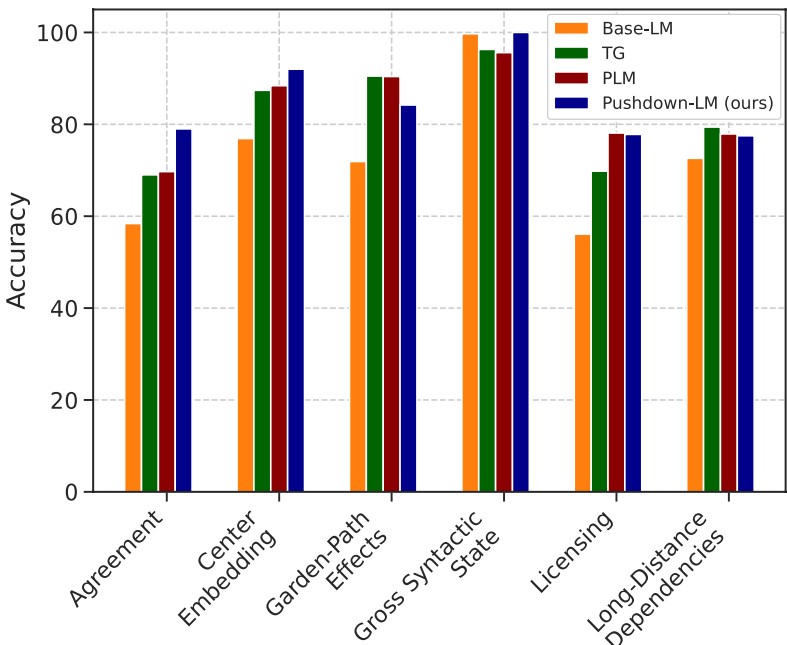

Figure 3: Comparing Pushdown-LMs with baseline Transformer LMs and other syntactic LMs. While Pushdown-LMs are comparable with Transformer Grammars (TG; Sartran et al., 2022) across all examples in SG test suites (Table 2), they outperform TGs on 4 out of 6 tests, including the recursive center embedding tests.

## 4.2 Sentence-Level Language Modeling

**Setup.** Next, we train 16-layer Pushdown Transformer LMs on the BLLIP-LG dataset of Charniak et al. (2000), with training splits from Hu et al. (2020), and the same pre-processing as Qian et al. (2021). We use the same hyperparameters (model size, dropout, learning rate schedulers) as Sartran et al. (2022). To measure syntactic generalization, we evaluate on BLIMP (Warstadt et al., 2020) and the SG test suites (Hu et al., 2020). In BLIMP, models are provided with a grammatical and ungrammatical sentence, and evaluated on assigning a higher probability to the grammatical sentence. SG test suites consist of an extensive set of handcrafted test cases, covering 6 fine-grained syntactic phenomena. Each test case involves satisfying a specific inequality constraint among surprisal values of various continuations given prefixes, where these inequalities are grounded in theories of incremental language processing—for instance, assigning a higher surprisal to the last verb in *The painting that the artist deteriorated painted* vs. *The painting that the artist painted deteriorated.* For BLIMP, we obtain $p(x)$ by approximate marginalization via beam search. Since surprisal values $-\log p(x_t \mid x_{<t})$ in SG test suites are meant to reflect incremental sentence processing, we perform marginalization based on the beam state at time

step $t$. We fix the beam size at 300.

**Results.** We present results on SG test suites in Figure 3. As baselines, we compare against a standard 16 layer Transformer LM and prior structured models (TG, PLM) from Sartran et al. (2022). As expected, all models with an explicit notion of structure have much better syntactic generalization across all test suites. Next, we note that Pushdown-LM, a 16 layer Transformer LM with all self-attention blocks replaced with Pushdown Layers, outperforms prior approaches—Pushdown-LM beats TG on 4/6 tests and PLM on 3/6 tests with similar performance on licensing. Next, we present results (averaged across 3 seeds) on BLIMP as well as aggregate SG test suite results and perplexity on the BLLIP test set in Table 2. Here, we note that Pushdown-LM achieves better syntactic generalization than prior structured models (including the PLM-mask model from (Qian et al., 2021)) on BLIMP. Finally, we find that Pushdown-LM achieves slight gains in perplexity compared to Base-LM.

## 4.3 Language Modeling with WIKITREES

Can Pushdown Layers continue to offer improvements on larger-scale language modeling? We construct WIKITREES, a dataset of over 100 million tokens extracted from Wikipedia Articles (WikiText-

| Model | BLIMP ↑ | SG test suites ↑ | PPL ↓ |
|---|---|---|---|
| **Models that add structural tokens to inputs** | | | |
| PLM | 75.1 | 80.2 | 29.8[‡] |
| PLM-Mask | 75.3 | 78.3 | 49.1[‡] |
| TG | – | **82.5**[*] | 30.3[‡] |
| **Models that do not add extra tokens to inputs** | | | |
| Base-LM | 70.1 | 69.5 | 20.1 |
| Pushdown-LM (ours) | **75.6** | 82.3[*] | **19.9** |

Table 2: Syntactic Generalization on BLIMP and SG test suites. All results for PLM-Mask are taken from Qian et al. (2021) and results for PLM and TGs are taken from Sartran et al. (2022). ∗ denotes differences that are not significant. PPL results marked with ‡ are taken from prior work and not comparable due to differences in tokenization.

103; Merity et al. (2017)), parsed automatically using a state-of-the-art neural constituency parser (Kitaev et al., 2019). Typically, LMs trained on web-scale data are given multi-sentence contexts with large window sizes as inputs, and to adapt this to Pushdown-LMs we make a small number of modifications (see Appendix B for details).

**Sample-Efficient Generalization.** To measure sample efficiency in Pushdown Transformers, we train LMs on [10M, 50M, 100M] tokens from WIK-ITREES. To ensure stable training under low data regimes, we train a 12 layer GPT2 using the exact configuration and tokenization scheme as GPT2-small (Radford et al., 2019), and additionally use dropout to prevent overfitting. For these experiments, we compare Base-LM with an LM where the final 6 self-attention blocks are Pushdown Layers (Pushdown-LM). To measure syntactic generalization, we compute aggregate performance on the SG test suites. From results in Fig. 4, we find that Pushdown-LMs exhibit drastically more sample-efficient syntactic generalization—for instance, syntactic generalization of Pushdown-LM trained on 10M tokens requires over 40M tokens for the Base-LM to surpass.

**Finetuning for text classification.** Can Pushdown Layers offer improvements on language understanding tasks, beyond syntactic generalization? To answer this, we perform staged finetuning of GPT2-medium with Pushdown Layers. Specifically, we finetune GPT-2 medium with the final 12 self-attention blocks replaced with Pushdown Layers (Pushdown-GPT2), as a language model on

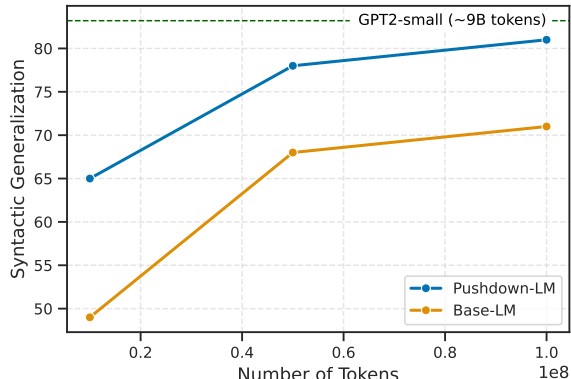

Figure 4: Comparing a standard GPT-2 small architecture (Base-LM) with a model where the last 6 self-attention blocks use Pushdown Layers, trained on various amounts of tokens from WIKITREES. We find that Pushdown Layers greatly improve sample efficiency of syntactic generalization. For reference, we also include GPT2-small, which is trained on over 9 billion tokens.

WIKITREES. We use this model to obtain parses on 4 text classification tasks: RTE, SST5, MRPC and STS-B from GLUE (Wang et al., 2019a), and use these parses to pre-compute the stack tape for every token. Then, in a second finetuning step, Pushdown-GPT2 is trained to perform text classification over these datasets by reducing each task into language modeling via prompting (See Appendix A for prompt details). As a comparison, we also perform the same staged finetuning for the standard GPT2-medium architecture. We report averaged results across 3 seeds in Table 3. We find that Pushdown Layers offer improvements on 3 out of 4 text classification tasks.

| Model | RTE | SST5 | MRPC | STS-B |
|---|---|---|---|---|
| GPT2 | 72.2 | **54.8** | 88.4 | 89.6/89.8 |
| Pushdown-GPT2 | **72.9** | 54.5 | **89.3** | **89.8/90.1** |

Table 3: Finetuning models on various semantic text classification/regression tasks. We report accuracy for RTE and SST5, F1 for MRPC, and Spearman/Pearson Correlation for STS-B.

## 5 Analysis

For all analyses, we use the 16 layer Pushdown-LM trained on BLLIP-LG from Section 4.2.

**Parsing.** Since Pushdown-LM is a syntactic language model, we obtain parses via beam search (beam size = 300) to approximately recover the most likely parse $y^* = \arg\max_y p(x, y)$ under our

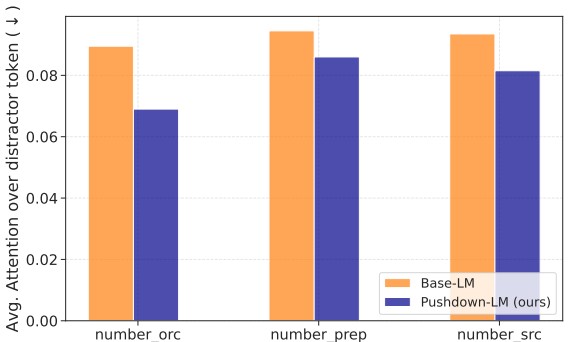

Figure 5: For the three subject-verb agreement tasks from (Marvin and Linzen, 2018), we compute average attention over the distractor noun when the verb is being predicted, for both the Base-LM and Pushdown-LM (ours). Across all variants, we find that our model consistently pulls attention away from distractor nouns.

model. However, since this parse is (a) unlabeled and (b) binarized, we perform an *unlabeled F1 evaluation* (using EVALB; Collins, 1997) over *binarized* ground-truth parses from the PTB test set. We also remove instances consisting of unknown words for our model, since our model is trained without any UNK tokens, giving us 2335 out of 2416 sentences. We compare our model against Kitaev et al. (2019), the parser that was used to annotate training data for Pushdown-LM. We also present unlabeled F1 on the auto-parsed BLLIP-LG test set. From results in Table 4, we note that our model achieves a very competitive unlabeled F1 score of 95.3, outperforming the official implementation of Kitaev et al. (2019)[3]. We also find that our model obtains a high F1 score of 97.3 on the BLLIP-LG test set.

| Model | PTB | BLLIP-LG |
|---|---|---|
| Pushdown-LM | **95.3** | 97.3 |
| (Kitaev et al., 2019) | 94.7 | - |

Table 4: Unlabeled F1 scores against binarized ground-truth parses from the PTB and BLLIP test sets. We filter all examples from the PTB test set with unknown words, giving us 2335 out of 2416 sentences. Annotations on BLLIP-LG are obtained using Kitaev et al. (2019).

**Case Study: Analyzing attention patterns on subject-verb agreement tasks.** We consider the 3 Subject-Verb agreement tasks (Marvin and Linzen, 2018) from the SG test suites. On these

tasks, models are presented with a prefix consisting of a main subject and a distractor embedded subject, where these items conflict in number. The objective is to assign a higher logprob to the verb that agrees with the main subject rather than the distractor subject. For instance, for prefix *The author that hurt the senators*, the model must assign a higher probability to *is* than *are*.

From Fig. 3, we find that Pushdown-LM significantly outperforms other models with close to 80% accuracy while Base-LM achieves less than 60% accuracy. To understand how Pushdown Layers modulate attention on these examples, we obtain attention scores over all prefix tokens (averaged across all layers). We present the average attention assigned to the distractor token for both Pushdown-LM and Base-LM in Fig. 5 where we observe that Pushdown-LM pulls attention away from the distractor noun, allowing it to predict the correct verb. Finally, we plot some (averaged) attention heatmaps in Fig. 6.

## 6 Other Related Work

While recursive structure is fundamental to natural language, modeling such structure is difficult for self-attention. Hahn (2020) considers DYCK, the simplest formal language with recursive structure, proving that hard attention cannot recognize DYCK and soft attention cannot recognize DYCK with low cross-entropy. In practice, we find that even simpler languages like PARITY are challenging for encoder-only Transformers (Chiang and Cholak, 2022; Bhattamishra et al., 2020). On the other hand, Transformers with decoders have been shown to be Turing-complete (Perez et al., 2021), but these constructions rely on the impractical assumption of running the decoder for an unbounded number of steps. In practice, we find that Transformer LMs struggle with generalization beyond regular languages and tend to learn shortcuts instead (Deletang et al., 2023; Liu et al., 2023).

Given these limitations, there is significant interest in inductive biases that encourage recursive structure in Transformers. One line of work considers constraining self-attention patterns according to syntactic parses (Strubell et al., 2018; Wang et al., 2019b; Peng et al., 2019; Deshpande and Narasimhan, 2020, among others). A second line of work adds structure to language modeling by learning joint probabilistic modeling of structure and strings (Chelba, 1997; Mirowski and Vlachos,

---

[3]We use the `benepar_en_large` model from `https://github.com/nikitakit/self-attentive-parser` which reports a score of 96.29 on the full PTB test set, while we obtain 95.66 (labeled F1, using the standard EVALB script).

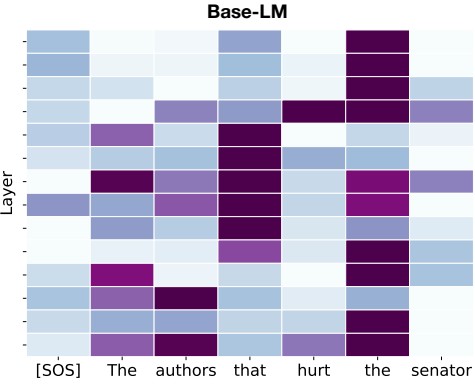

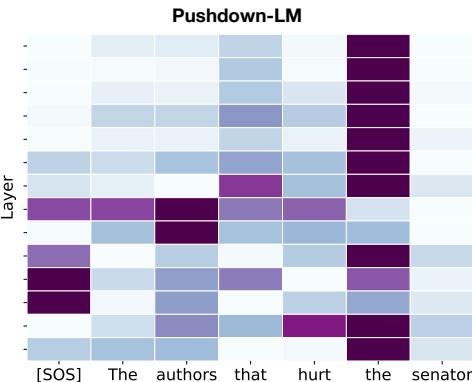

Figure 6: Given a prefix containing a main noun and a distractor noun, Pushdown-LM pulls attention away from the distractor (here *senator*), helping the model predict the verb with the correct number. These attention maps average across all the instances in the number_src test of SG test suites, and we show the attention over all prefix tokens when the main verb is predicted

2015; Choe and Charniak, 2016; Dyer et al., 2016, among others). Both of these ideas are combined in recent work of Qian et al. (2021); Sartran et al. (2022), that proposes joint string, parse Transformer language models with constrained attention patterns. While Pushdown Layers are also in this modeling tradition, we do so without operating on long transition actions, and enforce structural constraints via gradient based learning.

A separate line of work proposes neural networks augmented with structured memory like stacks (Das et al., 1992; Grefenstette et al., 2015; Joulin and Mikolov, 2015; DuSell and Chiang, 2022) or random access memories (Kurach et al., 2015). Such augmented neural networks are vastly better at algorithmic generalization and learning recursive structure (Suzgun et al., 2019; Deletang et al., 2023). Our work is the first that designs a structured memory (the stack-tape) for Transformers, that is updated just like stacks in a shift/reduce

manner, but unlike prior work, the specific design of Pushdown Layers makes training parallelizable.

Finally, there have been several efforts to add syntactic inductive biases into sequence models (typically RNNs) that can acquire and use parse structures in an unsupervised manner (Bowman et al., 2016; Shen et al., 2019; Drozdov et al., 2019; Kim et al., 2019, among others). We leave unsupervised training of Pushdown Transformers for future work.

# 7 Conclusion

We propose Pushdown Layers, a new kind of self-attention that augments Transformer language models with a stack based memory. Pushdown Layers enable auto-regressive Transformers to softly bias attention towards a recursive syntactic computation, through an updatable stack-tape that stores token depths in an incremental parse. When trained on synthetic and natural languages, we find that Transformer LMs with Pushdown Layers achieve better generalization to deep recursive structure, as well as better and more sample-efficient syntactic generalization. When pre-trained LMs are finetuned with Pushdown Layers, we obtain improvements on some GLUE tasks.

# 8 Reproducibility

Code and data for these experiments is available at https://github.com/MurtyShikhar/Pushdown-Layers.

# Limitations

Pushdown Layers require constituency-parse annotated datasets, which may not be available for many languages due to a lack of high performing off-the-shelf constituency parsers. This also limits applicability to domains beyond natural and synthetic languages, such as algorithmic reasoning. Finally, Pushdown Layers can only be applied to languages with constituency structure, and our experiments are limited to English.

# Acknowledgements

SM was funded by a gift from Apple Inc. CM is a fellow in the CIFAR Learning in Machines and Brains program. PS and JA are funded by Project CETI via grants from The Audacious Project: a collaborative funding initiative housed at TED. We thank John Hewitt, Sidd Karamcheti and Róbert Csordás for feedback and discussions.

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

## A  Model Hyperparameters

**BLLIP.**   All our hyperparameters for BLLIP are borrowed from the 16-layer Transformer LM used in Sartran et al. (2022). This includes model hyperparameters (hidden state dimension, number of attention heads, number of layers), dropout (input dropout, output dropout, attention dropout), and learning rate schedulers. We train for 300k steps, evaluating every 3k steps and early stop based on validation set perplexity.

**WikiTrees.**   For experiments on WikiTrees, we use the same model hyperparameters as GPT2-small, and a context window of 512. We train with a batch size of 480, and train till validation loss stops decreasing, with a learning rate that linearly warms up from 0 to 6e-4 over 200 iterations, followed by a cosine learning rate scheduler. For sample efficiency experiments, we add dropout of 0.2 to prevent overfitting.

**GPT2-medium finetuning.**   We use a batch size of 256, and a constant learning rate of 3e-5. We early stop based on validation set performance, and report average of 3 runs. To convert text classification tasks into language modeling, we use the following prompts:

- **RTE:** *Premise: {p}. Hypothesis: {h}. Label: {l}*, given a premise, hypothesis pair $(p,h)$ with label $l$ mapped into {Yes, No}.

- **MRPC:** Given the sentence pair $(s_1, s_2)$, we create a prompt *Sentence1: {s_1}. Sentence2: {s_2}. Label: {l}.* where $l \in \{0, 1\}$.

- **SST5:** *Sentence: {s}. Sentiment: {l}* for an input sentence $s$ with label $l$.

- **STS-B**: Given the sentence pair $(s_1, s_2)$, we create a prompt *Sentence1: {s_1}. Sentence2: {s_2}*, and use the final hidden state to featurize a linear regressor, trained jointly with the LM.

## B  Training Pushdown-LM with context-windows

In standard LMs, context windows for training are arbitrary offsets into the entire corpora—a window might start in the middle of some sentence. Because Pushdown-LMs always start with the stack state initialized as all 0s and make attachments only to stack tape contents, a Pushdown-LM cannot start in the middle of a sentence without the stack tape

appropriately initialized. We get around this by simply sampling these context windows to always start at sentence boundaries. We also prepend a special token ROOT before the start of every sentence such that the attachment decision of the final word is made to this ROOT token.

## C  Parsing with Pushdown-LM.

Since the BLLIP-LG trained Pushdown-LM operates over sub-word tokens, parses produced by this model have subwords as leaf nodes. We process this by recursively merging leaf siblings that are part of the same word. For instance, given the bracketing *(ab, (ra, (ca, dabra)))*, we recursively merge these to get a single node *abracadabra*. This procedure deterministically converts the parse over subwords into a parse tree over words.

## D  Implementation details: Pseudocode for implementing Pushdown Layers

See Fig. 7 and Fig. 8 for reference implementations of Pushdown Layers and the attachment head.

```python
class PushdownSelfAttention(nn.Module):
    ...

    def forward(self, x, stack_tape):
        B, T, C = x.size()
        q, k, v = self.c_attn(x).split(self.n_embd, dim=2)

        # (B, nh, T (dest), hs)
        q = q.view(B, T, self.n_head, C // self.n_head).transpose(1, 2)
        # (B, nh, T (src), hs)
        k = k.view(B, T, self.n_head, C // self.n_head).transpose(1, 2)
        # (B, nh, T, hs)
        v = v.view(B, T, self.n_head, C // self.n_head).transpose(1, 2)

        augmented_keys = k.unsqueeze(2) + self.beta(stack_tape.int()).unsqueeze(1)
        augmented_keys /= math.sqrt(k.size(-1))
         # B x nh x T (dest) x T(src)
        augmented_att = (q.unsqueeze(3) @ augmented_keys.transpose(-2, -1)).squeeze
            ↪ (3)

        att = augmented_att.masked_fill(self.bias[:, :, :T, :T] == 0, float("-inf"))

        att = F.softmax(att, dim=-1)
        att = self.attn_dropout(att)
        # (B, nh, T, T) x (B, nh, T, hs) -> (B, nh, T, hs)
        y = att @ v
        # re-assemble all head outputs side by side
        y = y.transpose(1, 2).contiguous().view(B, T, C)
        # output projection
        y = self.resid_dropout(self.c_proj(y))
        return y
```

Figure 7: Python implementation of a Pushdown Layer attention block.

```
1   class AttachmentHead(nn.Module):
2       ...
3       def forward(self, x, stack_tape, next_word):
4           B, T, C = x.size()
5
6           q, k = self.data_to_qk(x).split(self.embd_dim, dim=2)
7           # (B, T (dest), hs)
8           next_word_q = self.q_next_word_mlp(torch.cat([q, next_word], dim=-1))
9           # (B, T (dest), hs)
10          next_word_k = self.k_next_word_mlp(torch.cat([q, next_word], dim=-1))
11          # (B, T (dest), T (src), hs)
12          k = k.unsqueeze(1).repeat(1, T, 1, 1)
13          # B x  T (dest) x T (src) x hs
14          depth_embds = self.beta(stack_tape.int())
15          k_with_write_info = self.key_and_stack_mlp(torch.cat([k, depth_embds], dim
                ↪ =-1))
16
17          # first, calculate attention score between query and keys
18          k_with_write_info /= math.sqrt(k.size(-1))
19          attach_logits = (next_word_q.unsqueeze(2) @ k_with_write_info.transpose(-2,
                ↪ -1)).squeeze(2)
20
21          # if no reduce, then we compute score with itself
22          next_word_k /= math.sqrt(k.size(-1))
23          logits_self = (next_word_q.unsqueeze(2) @ next_word_k.unsqueeze(3)).squeeze
                ↪ (2)
24
25          # now insert logits_self into the k+1th position of attach_logits for each k
26          pad_tensor = torch.zeros(attach_logits.shape[0], attach_logits.shape[1], 1,
                ↪ device=attach_logits.device)
27
28          attach_logits_l = torch.cat([attach_logits, pad_tensor], dim=-1)
29
30          logits = attach_logits_l.scatter(
31              2,
32              (1 + torch.arange(T))
33              .unsqueeze(0)
34              .unsqueeze(-1)
35              .repeat(B, 1, 1)
36              .to(attach_logits_l.device),
37              logits_self,
38          )
39
40          # B x T x T+1. => B x T+1 x T+1 by padding the first row with zeros
41          logits = torch.cat(
42              [
43                  torch.zeros(
44                      logits.shape[0],
45                      1,
46                      logits.shape[2],
47                      device=logits.device,
48                  ),
49                  logits,
50              ],
51              dim=1,
52          )
53
54          # set upper triangular part to -inf
55          logits = logits.masked_fill(self.bias[:,:T+1, :T+1] == 0, float("-inf"))
56          return logits[:, 1:]
```

Figure 8: Python implementation of the Attachment head in Pushdown Transformers.