# OpenReview forum: "Pushdown Layers: Encoding Recursive Structure in Transformer Language Models"
_EMNLP/2023/Conference — EMNLP 2023 Main_

### Official Review · Reviewer_tLfe · 2023-08-01

**Soundness:** 4

**Excitement:**

4: Strong: This paper deepens the understanding of some phenomenon or lowers the barriers to an existing research direction.

**Missing References:**

More types of neural networks augmented with stacks to put in Section 6:

* ON-LSTM https://arxiv.org/abs/1810.09536
* Nondeterministic Stack RNN https://arxiv.org/abs/2210.01343


**Paper Topic And Main Contributions:**

This paper presents a modification of the transformer architecture, called $\lambda$-layers, designed to model recursive syntax. The method requires syntactic supervision during training and can be used to extract unlabeled, binarized parse trees. The method works by inserting a new mechanism just prior to the attention layer that resembles a shift-reduce parser. While reading the input, it keeps track of a stack of constituents, and at each timestep, it decides how many constituents will be reduced with the next token. The output of this mechanism is the depth of each input token, i.e. the number of reduce operations it has been involved in. These depths are added to the keys of the standard attention layer, so the model can attend to inputs based on the structure of the syntax tree.

The authors compare this method against a baseline transformer that is trained with the same syntactic information using multi-task learning, and other recent transformer architectures that jointly model sequences and parses: Transformer Grammars (TGs) by Sartran et al. (2022), and PLM by Qian et al. (2021). The main experimental results are:
1. On the Dyck language, $\lambda$-layers generalize better than baseline transformers
2. Mixed results on the Syntactic Generalization test suites (Hu et al. 2020), but always better than the baseline transformer
3. Better BLIMP score than baseline transformers
4. Slightly better perplexity than baseline transformers on the BLLIP test set
5. Much better data-efficiency than the baseline GPT2 architecture on a large language modeling task, evaluating on SG score
6. Mixed results on GLUE, but often better
7. Better unlabeled, binarized parsing on BLLIP than the model of Kitaev et al. (2019)
8. Analysis of the learned attention patterns on a subject-verb agreement task with distractor noun phrases (Marvin and Linzen 2018)

Overall, it looks like $\lambda$-layers are better than baseline transformers at learning recursive syntax and are at least competitive with existing syntactically-supervised baselines.


**Questions For The Authors:**

A. 087, 185: It's not clear what you mean by fewer constraints on attention mechanisms. In what way are $\lambda$-layers "softer" and more general in relation to TGs and PLM?

B. $W_k$ keeps track of the depth of each token, but is there anything that lets the model detect the boundaries of constituents with the same depth? For example, in Fig 2, is the model able to tell that at $t = 2$, "cat" and "is" are in separate constituents?

C. When the lambda layer selects token $x_i$ to reduce with out of $\{ x_1, \ldots, x_k \}$, does this mean that $x_i, ..., x_k$ are reduced into a new constituent?

D. What does it mean when $x_k$ is reduced with itself ($r_k = x_k$)?

E. Why is the "on-the-fly" vector for timestep $k$ computed from $h^L_{k-1}$, as opposed to any other $h^L_i$? There's no recurrence, so I don't see what would be special about the last timestep.

F. 248: If attachment decisions are made using the score for the rightmost token in the constituent, what happens if the model selects a token in the middle of a constituent? For example, if selecting "cat" causes the (The cat) to be part of the new constituent, what would happen if "The" were selected?

G. 270, footnote 1: What is the overall time and space complexity of the $\lambda$-layer? Is this an *extra* factor of $n$, making it $O(n^3)$?

H. 271: What objective function is used for attachment probabilities, and how is it weighted vs. the LM objective?

I. This seems like a fair comparison, although I would be interested in more details about the multi-task setup.

J. 302: Do you exclude opening brackets from this prediction, and just take the highest of the closing brackets?

K. Table 1: How many examples are used in the test set? Do you evaluate on different positions within the same Dyck string?

L. Table 1: Why do you think your model doesn't get 100%?

M. 301: Can you give more details of how these prefixes are generated? For example, for the depth gen. task, is the prefix just a sequence of 15-50 open brackets?

N. 295: Can you give more details of the distribution over strings in the training data? What is your algorithm for sampling Dyck strings?

O. 293: Do all 6 layers have $\lambda$-layers?

P. 351: The results on SG score look pretty mixed for $\lambda$-LM. Can you explain why $\lambda$-LM might be doing well on some test suites but not on others? These tests are of quite different character. Garden-Path Effects and Center Embedding are arguably the most syntax-oriented.

Q. Table 2: I would be curious to see results for Base-LM without being trained with attachment data, but this isn't required. I ask because I'm curious how much of a boost you get just by adding the attachment data.

R. 409: For the GPT2-medium architecture, do you also train with attachment decisions using multi-task learning?

S. What is the expected effect of stacking multiple $\lambda$-layers? Shouldn't just one be enough to implement a shift-reduce parser?

T. Why use different, unrelated embeddings for each of the depths? What if you just used one embedding for depth 0 and the same embedding for all depths >0?

U. Fig 6: $\lambda$-LM still doesn't seem to attend to "authors" much. Mostly it attends to "the". Any idea why that happens?


**Reasons To Accept:**

This paper makes progress on an important research question: how can we modify the transformer architecture so that it learns recursive syntax more effectively? The authors present a new model and show that it consistently outperforms the baseline transformer, and it performs at least competitively with two recent baselines that similarly model sequences and parses jointly. Unlike these baseline methods, this paper's method does not require a "bespoke" decoding procedure because the syntactic structure is not part of the output sequence.

The experimental section of the paper includes an excellent variety of tasks, evaluation metrics, and baseline models. A particularly nice result IMO is that $\lambda$-layers are much more data-efficient that baseline transformers in terms of SG score (see line 289).


**Reasons To Reject:**

As pointed out in the paper, there are already a couple of syntactically-supervised transformer architectures out there, and $\lambda$-layers don't seem to have a decisive advantage in terms of task performance.

I found the description of some aspects of the method to be vague, and I would appreciate more details. See the Questions section for specifics. Something that would be beneficial for understanding the method is a plain-English description of what Algorithm 1 is doing.

I didn't find the conclusions about the learned attention patterns from Fig. 6 very convincing (see question U).


**Reproducibility:**

3: Could reproduce the results with some difficulty. The settings of parameters are underspecified or subjectively determined; the training/evaluation data are not widely available.

**Reviewer Confidence:**

4: Quite sure. I tried to check the important points carefully. It's unlikely, though conceivable, that I missed something that should affect my ratings.

**Typos Grammar Style And Presentation Improvements:**

A. Can you explain why you chose the name $\lambda$-layer?

B. Is "depth" really referring to the height of the tree of each reduced constituent? In a shift-reduce parser, "depth" would refer to the depth of the stack of shifted terminals and nonterminals.

C. 208: A *deterministic* PDA.

D. 260: What does "arbitrarily non-linear" mean?

E. 266: It would be good to describe how to precompute these ground-truth values from the parses.

F. 287: I don't think this has to do with the fact that they're predicted in parallel (which they aren't?), but with the fact that the output vocabulary does not include brackets for encoding the parse tree.

G. Fig 6: Could you label the layers on the y axis? I don't know which direction the layers go in.

---

> ### Author Rebuttal · Authors · 2023-08-29
>
> Thank you for the very thoughtful and comprehensive review! We are glad you liked the variety of experiments.
>
>
> **Decisive advantage over other approaches?**
>
> We argue that our method is better by not tripling the sequence length with structural tokens, and not needing a bespoke decoding procedure. We decisively are better than PLM and PLM-mask models of Qian et al. as shown in Table-2. Next, compared to transformer grammars, we suggest that we do win on more syntactic tests (as shown in figure 3), but nevertheless we do accept that the quantitative gains are present but not decisive.
>
>
>
> **Attention patterns in Fig 6?**
>
> Note that since the model is auto-regressive, information about the prefix gets automatically propagated into the next word. Thus, information about the word “authors” is already present in “that”, “hurt” and “the”. However, information about the distractor word “senator” is not present in any of the other words, and the only way that information is used is via self-attention. Thus, a model that has higher attention over the distractor is likely to predict the incorrect verb. On the other hand, a model need not attend over the main noun (authors) to predict the correct word since information about this word is also contained in “that”, “hurt” and “the”.
>
> Here, we see that both models read off information about the main noun from the word the but lambda-LM assigns smaller attention weight to the word *senator*, indicating that it is less affected by this distractor word.
>
> Thank you for the additional references and presentation suggestions, which we will use to improve the paper.
>
> **Answers to questions:**
>
> A) TGs impose a hard attention constraint — once a constituent closes, subsequent words are not allowed to attend to words inside the constituent, and can only attend to the special constituent token. Lambda-layers have no such constraint, and the model can learn to attend to any word in the prefix.
>
> B) No, but since the stack information is updated autoregressively, the stack contents at the previous time step can inform the model about constituent boundaries.
>
> C) Yes.
>
> D) This corresponds to a “shift” operation. We will annotate Eq 5 so it indicates this.
>
> E) We use this vector to represent the next word because it’s closest in vector space to the next word (since it will be part of a softmax and trained to predict the next word). All of the other $h_i^{L}$ will be closer in vector space to word $w_i$.
>
> F) We ensure that this never happens i.e. once a constituent is formed, only the rightmost word of that constituent is allowed to be chosen (can be easily implemented by setting scores for middle tokens to -INF), see Fig-2 caption and the note in Line 244.
>
> G) Time complexity is the same as ordinary self-attention. At training time, instead of keeping 2D tensors, we have to store 3D tensors for computing attention scores (since depth of a prefix token changes multiple times as a sentence as processed) so the memory requirement is $O(bn^2)$ where b is the max sentence length across all sentences in the input string. For inference, memory requirement is the same as ordinary attention.
>
> H) It’s a multi-class classification loss based on probabilities from Eq 5 (i.e. maximize probability of correct attachment). Classes are weighted equally.
>
> I) We use the same multi-task setup as the Scaffold-LM model from https://aclanthology.org/2021.acl-long.289v2.pdf.
>
> J) Yes, this prevents opening brackets from competing.
>
> K) Yes, 5000 examples in the test set (we will include these details in the appendix).
>
> L) Since the model has to infer the parse structure at test time, it can fail to do so for some strings, resulting in incorrect predictions.
>
> M) We sample a dyck string of a given length, and then prepend an open bracket. Then, we measure if the model can close this bracket. i.e. for length 5, we might sample `()()` and prepend `[`, giving `[ () ()`.
>
> N) We just sample uniformly from the language (using an implementation similar to https://github.com/princeton-nlp/dyck-transformer).
>
> O) Yes.
>
> P) Note that we outperform all methods on center embedding.
>
> Q) Sure, happy to include this. For BLIMP, we find that without the multi-task loss, the base-LM gets 67.8% (compared to 70.1) and on SG test suites, we find that the model gets about 67% (compared to 69.5%).
>
> R) Yes, which is why it gets close to ~69 with 50M tokens (similar to the size of BLLIP-LG).
>
> S) This is true, in theory but in practice, multiple stacked lambda-layers improved performance.
>
> T) This simply provides more features to the model as opposed to just knowing if a token has been closed or not.
>
> U) Please see our explanation for figure-6 above.

---

### Official Review · Reviewer_iyy7 · 2023-08-04

**Soundness:** 3

**Excitement:**

4: Strong: This paper deepens the understanding of some phenomenon or lowers the barriers to an existing research direction.

**Missing References:**

56, 462: Yao et al actually show that transformers can model unbounded recursive structure (Appendix B.4), but express doubt about how well it can be learned.

470: Chiang and Cholak (2022) actually show that transformers can recognize PARITY, although, again, they express doubt about how well it can be learned.

499: Further work on neural networks with stacks has been done since 2015:
- https://arxiv.org/abs/1911.03329
- https://ieeexplore.ieee.org/ielaam/9078688/9445210/9337907-aam.pdf?tag=1
- https://arxiv.org/pdf/2109.01982.pdf
- contemporaneous work: https://arxiv.org/pdf/2304.12955.pdf chapter 10, which uses stacks in transformer layers


**Paper Topic And Main Contributions:**

This paper describes \lambda-LM, a language model using \lambda-layers, which perform a shift-reduce parse, whose stack history is used to influence self-attention. It is trained on gold or "silver" trees. Experiments are on both synthetic languages (Dyck) and various natural language benchmarks for syntactic generalization.


**Questions For The Authors:**

A. Why are \lambda-layers called that? They remind me strongly of \lambda-calculus, and I was confused when I discovered that your model has no lambdas in it. I'd strongly recommend a different name.

Section 3.1: The presentation of the model is not very precise.

B. There are a number of words written in scare quotes that should be defined better: "attachment decision", "recursive state", "on-the-fly",

C. It talks about shift/reduce operations but doesn't define clearly what grammar these operations use. Is the grammar just S -> S S, S -> a for all words a?

D. Does the stack tape losslessly encode a partial run of a shift-reduce PDA?

E. Why are reduces described as being between two tokens? Shouldn't they be between two constituents (which could also be tokens)? Can x_k be reduced with a token that is at depth > 0? Presumably this has something to do with the note at line 244, but it's not clear.

F. Why is there a nonzero probability for x_k to reduce with itself? Is that a shift? Please explain.

In Figure 2,

G. Why are constituents drawn as left-pointing arrows, instead of `/\` as is standard?

H. What do solid, dotted, and red lines mean?

I. At time t=1, what is the significance of the attachment head pointing just slightly to the right of the 1?

Section 3.3

J. Do you have to do anything special to backpropagate through the argmax?


**Reasons To Accept:**

The model is interesting, and the experiments are thorough with good results.


**Reasons To Reject:**

The description of the model in Section 3 is not very precise. It uses vague terminology in scare quotes, and it seems to make assumptions about how a shift-reduce parser works that might be particular to dependency parsing (?) and not shared by everyone.

Not reasons to reject, but weaknesses that I think should be corrected:

Section 4: For all of these tasks, it seems to me there's more than one way to construct the binary-branching trees that are used in training. Could you please add explanations?

The requirement of gold or "silver" trees is a hard requirement, but I feel the wording of the paper doesn't quite convey this. It says "When trained on a corpus of strings annotated with silver constituency parses" (17) and it "can be trained on any text corpus annotated with constituency parses" (80), which leaves open the possibility that it can be trained on unannotated text. I think you could be more up-front about this, for example, by calling \lambda-LM a "syntactic language model" or \lambda-layers "syntactic self-attention layers".


**Reproducibility:**

3: Could reproduce the results with some difficulty. The settings of parameters are underspecified or subjectively determined; the training/evaluation data are not widely available.

**Reviewer Confidence:**

4: Quite sure. I tried to check the important points carefully. It's unlikely, though conceivable, that I missed something that should affect my ratings.

---

> ### Author Rebuttal · Authors · 2023-08-29
>
> Thank you for the comprehensive and very helpful review! And we’re glad you found the model interesting.
>
> **The requirement of gold or "silver" trees is a hard requirement”**: Thank you for mentioning this. As you note, we do point it out early on, we actually use the term “syntactic language model” as a submission keyword, and we mention several times (line 77, 416) that lambda-layers model the joint probability of strings and parses. Nevertheless, we will reword to emphasize this unambiguously and further adopt the term syntactic language model.
>
> We will aim to make the model presentation in sections 3 and 4 much more precise. Thank you also for the helpful references and discussion of them. We believe that our model benefits from the helpful inductive bias rather than what can be represented.
>
> Next, we respond to questions:
>
> Questions about Section 3.1:
>
> A) We take inspiration from $\lambda$-calculus where constructions such as the Y combinator are used to implement recursion in programming languages that lack an implementation for recursion.
>
> B) **Defining Attachment decision**: attaching the current token with a prefix token j refers to constructing a constituent by performing multiple reduce operations with the top element of the stack until prefix token j is included in the constituent. This new constituent is then pushed on the stack. (Please see Algorithm 1 for a precise specification of this operation)
>
> C) **Defining the grammar**: Yes, we model a language of unlabeled brackets, so the grammar is is `S -> SS`, `S and S -> a`.
>
> D) **Does the stack-tape losslessly encode…?** Yes.
>
> E) **Why do reduce operations involve two tokens?** Reduction operations ultimately build up constituents (chosen from the incremental parse, see Lines 223-224). However, the module that predicts these decisions does so using a single summary token representation from each constituent. In particular, this is the right-most token: as we mention in the note on Line 244, this is necessary since in left-to-right LMs, only the rightmost token of a span contains all the information about words in that span.
>
> F) **Why can x_k reduce with itself?** Yes, reducing x_k with itself corresponds to a shift operation. We will annotate Eq 5 so that this is clearer.
>
> Questions about Figure 2:
>
> G) **Why arrows for constituents?** This is intended as a visual representation of how the constituent-building operation works: *cat* “attends” backward to reduce with *the*, and *happy* attends backward to reduce with *[the cat]*, which requires first reducing with *is*. We’ll clarify this in the figure caption.
>
> H) **What do lines mean?** We will clarify this in the figure caption. Solid line: the chosen token, according to Eq~5. Red line: token that cannot be chosen since it is not the rightmost token of any constituent. Dotted line: some candidate token that was not chosen.
>
> I) **Why does the attachment head point to the right of the 1?** Sorry, this is just a positioning issue. The arrow from the attachment head to the stack tape is meant to indicate that the attachment head outputs modify the *entire* stack tape (positioning w.r.t individual numbers in the tape is coincidental)
>
> J) **Backprop through argmax?** We only have to perform this inference at test time, since we use gold / silver trees to compute stack tape values for learning (Line 267). We think that treating these attachment decisions as latent is an exciting direction for future work.

---

### Official Review · Reviewer_HLRB · 2023-08-05

**Soundness:** 3

**Excitement:**

3: Ambivalent: It has merits (e.g., it reports state-of-the-art results, the idea is nice), but there are key weaknesses (e.g., it describes incremental work), and it can significantly benefit from another round of revision. However, I won't object to accepting it if my co-reviewers champion it.

**Missing References:**

Although this paper focuses on parse tasks, some similar ideas are explored to consider the recurrent (or recursive) mechanism for attention models (also the usage of embeddings for additional states in every layer), which can be adapted to parse tasks.

M. Dehghani, S. Gouws, O. Vinyals, J. Uszkoreit, and Ł. Kaiser, “Universal transformers,” in 7th International Conference on Learning Representations, ICLR 2019, 2019, pp. 1810–1822.

X. Ai and B. Fang, “Leveraging Relaxed Equilibrium by Lazy Transition for Sequence Modeling,” in Proceedings of the 60th Annual Meeting of the Association for Computational Linguistics, 2022, vol. 1, pp. 2904–2924.

**Paper Topic And Main Contributions:**

This paper studies the recurrent/recursive inductive bias (like RNN’s) for Transformer-based parse models. To track the depth,  the authors suggest a self-inferred stack tape that records the recursive state for each token. To update the stack tape record for the current token, the authors consider a model of probabilistic attachment decision to find the relevant prefixes, attach the current token to the relevant prefixes, and update the record of the current token in the stack tape based on the record of the relevant prefixes. Finally, to use the stack tape, in each  $\lambda$-layer, authors use records as embeddings to force recursions and add them to the layer input.

Experiments on 4 parse tasks support the idea.

**Questions For The Authors:**

Please see Reasons To Reject.1.

Table 2 confuses me. As the stack tape provides parse information, adding embeddings of the stack tape gives the model structural information. Why do the authors believe they are not structural information?  For instance, in Figure 1, the stack tape "1 1 0 0 1 2 2" gives structural information regardless of the numerical formation. Do you mean you are using learnable structural information while baseline models apply off-the-shelf POS labels?

**Reasons To Accept:**

1. This work is well-motivated, the idea is straightforward, and the method is simple.
2. The method shows advances in parsing long dependencies (Table 1).

**Reasons To Reject:**

1.  I do not understand why the authors believe $\lambda$-LM pulls attention away from the distractor in Figure 6. This figure shows that  $\lambda$-LM still pays attention to the distractor at the bottom layers, and base-LM has a similar behavior but at the middle layers. Since the authors add the embeddings of the stack tape to the layer input, the authors might provide more analysis about how the embeddings impact the attention behavior.

2. Following the above question, some import variants might be provided and discussed. The authors add embeddings of the stacked tape to self-attention's "q" and "k" (i.e., attention score). How about applying to "v"? Why do you not apply them to Eq.5? Does it imply that the depth does not impact the attachment?

I'm happy to discuss these questions.

**Reproducibility:**

3: Could reproduce the results with some difficulty. The settings of parameters are underspecified or subjectively determined; the training/evaluation data are not widely available.

**Reviewer Confidence:**

4: Quite sure. I tried to check the important points carefully. It's unlikely, though conceivable, that I missed something that should affect my ratings.

---

> ### Author Rebuttal · Authors · 2023-08-29
>
> Thank you for the review - we were pleased that you found $\lambda$-layers a well motivated and straightforward approach.
>
> **Does the lambda-LM really pull attention away from distractors**? In Fig 6, the distractor is the noun *senator* (as mentioned in the caption), and the relevant observation is that the lambda-LM places less mass on this token than the ordinary LM (compare the last column in the two sub-figures, especially in layers 11 and 8). We apologize if this figure is unclear! This can also be seen quantitatively in Fig 5, which aggregates these attention weights over the entire Marvin & Linzen dataset.
>
>
> **Why not modify values?** The intuition is that a token’s syntactic position should change which other tokens attend to it (via keys). However, syntactic position should not change a token representation’s *content* (via its value embedding). We agree that these other variants would be interesting to explore in future work. We’d be happy to discuss possibilities with you further during the discussion phase.
>
> **Questions about Table-2**: Table 2 divides models based on whether they add *structural tokens* to the input string. The distinction is not whether structural information is available (we agree that lambda-layers also provide structural information), but instead the cost of including this information. As mentioned in background (Lines 173-194), models in the top portion of the table increase the size of the input string by adding constituent labels i.e. an input string “The man likes apples” becomes “(S (NP The man NP) (VP likes apples VP) S)”. This increases input size from n tokens to ~3n tokens that must receive full forward passes (in contrast to lambda-layers, where structural computation takes place within an ordinary forward pass).

---

### Meta-Review · Area_Chair_czmf · 2023-09-18

**Recommendation:** 4

**Metareview:**

This paper improves Transformers by adding \lambda-layers, which perform a shift-reduce parser and enable the self-attention over the stack history (paring prefix) beside the surface tokens. Transformers with \lambda-layers are jointly trained on strings with annotated (or parsed) trees. Experiments on both synthetic languages and various NL benchmarks show the generalization of their approach. All reviewers raise some questions about the clarity of this paper and some missing references related to shift-reduce stack parsing, recursive mechanism over attentions.

---

### Decision · Program_Chairs · 2023-10-07

**Decision:**

Accept-Main

**Comment:**

This paper improves Transformers by adding \lambda-layers, which perform a shift-reduce parser and enable the self-attention over the stack history (paring prefix) beside the surface tokens. Transformers with \lambda-layers are jointly trained on strings with annotated (or parsed) trees. Experiments on both synthetic languages and various NL benchmarks show the generalization of their approach. All reviewers raise some questions about the clarity of this paper and some missing references related to shift-reduce stack parsing, recursive mechanism over attentions.